# A Realist Evaluation of Theory about Triggers for Doctors Choosing a Generalist or Specialist Medical Career

**DOI:** 10.3390/ijerph17228566

**Published:** 2020-11-18

**Authors:** Belinda O’Sullivan, Matthew McGrail, Tiana Gurney, Priya Martin

**Affiliations:** 1Faculty of Medicine, The University of Queensland, Rural Clinical School, Locked Bag 9009, Toowoomba 4350, DC Queensland, Australia; t.gurney@uq.edu.au (T.G.); priya.martin@uq.edu.au (P.M.); 2Faculty of Medicine, The University of Queensland Rural Clinical School, 78 on Canning St, Rockhampton 4700, Queensland, Australia; m.mcgrail@uq.edu.au

**Keywords:** career choice, generalist, general practice, specialist, medical training, doctors, realist evaluation, theory, experience, norms, attributes

## Abstract

There is a lack of theory about what drives choice to be a generalist or specialist doctor, an important issue in many countries for increasing primary/preventative care. We did a realist evaluation to develop a theory to inform what works for whom, when and in what contexts, to yield doctors’ choice to be a generalist or specialist. We interviewed 32 Australian doctors (graduates of a large university medical school) who had decided on a generalist (GP/public health) or specialist (all other specialties) career. They reflected on their personal responses to experiences at different times to stimulate their choice. Theory was refined and confirmed by testing it with 17 additional doctors of various specialties/career stages and by referring to wider literature. Our final theory showed the decision involved multi-level contextual factors intersecting with eight triggers to produce either a specialist or generalist choice. Both clinical and place-based exposures, as well as attributes, skills, norms and status of different fields affected choice. This occurred relative to the interests and expectations of different doctors, including their values for professional, socio-economic and lifestyle rewards, often intersecting with issues like gender and life stage. Applying this theory, it is possible to tailor selection and ongoing exposures to yield more generalists.

## 1. Introduction

Many countries are training more doctors than ever before, but a major goal is achieving enough generalists working in fields like general practice (GP) and public health compared with narrow specialty fields [1]. Achieving a critical mass of generalists is important as they support delivery of integrated, preventative and primary care services across a wide range of community needs, at lower cost, for increased life expectancy [2,3,4,5]. Although preventive and primary care services are universally needed, many countries are facing declining general practice numbers [6,7]. Current trends are producing an overabundance of non-GP specialists who focus on targeted populations or body systems, potentially increasing the geographic centralisation (city practitioners), fragmentation and inefficiencies of healthcare. A more generalist workforce could be realised if the levers underpinning the choice to become a generalist or specialist doctor, were better understood.

The existing evidence of specialty choice is limited to countries where there are strong markets for specialist services, including Australia, the United Kingdom (UK), the United States of America (USA), Canada, Germany and Japan. Some is based on medical student intentions [8,9,10,11,12,13,14], somewhat unreliable for informing actual choice. Other material explores preferences of junior (pre-registrar) doctors [15,16,17,18,19,20,21] or trainees (registrars/residents enrolled in postgraduate vocational training) along with qualified fellows (generalist/specialist) [22,23,24,25,26]. However, this evidence is largely analysed by influential factors, not specifically about how these factors are activated (including for whom and when various choices might fire), which would better inform the design of interventions to produce generalists, across the long medical training pathway.

The literature highlights that choosing a specialty is a complex process with a number of identified correlates. One national survey of trainees suggested choice of a particular specialty was stimulated by *intrinsic*—appraisal of skills against specialty; intellectual content; interest in helping people; and *extrinsic* factors—work culture; flexible working hours and; hours of work [26]. Compared with other specialties, general practice trainees showed a higher regard for helping people and fitting their work to domestic circumstances [26]. General practice is also attractive because of lifestyle, continuity of care, procedural skills and work opportunities [15,16]. Primary care role models and experiences may facilitate uptake of general practice [15,24,27], although scant studies suggest general practice may have lower professional status compared with focused specialties [15,16,24]. Higher professional status is attributed to specialty fields like surgery that give a clear professional identity and tight network of inherent socio-economic capital [28]. 

Particular specialties may also be attractive to young and emerging doctors because of their pro-social attributes, like teamwork and caring, which reinforce expected values, norms and cultures [28]. Equally technical attributes may be a drawcard. Cardiology [25], surgery, obstetrics and gynaecology, ophthalmology, anaesthesia and emergency medicine, were attractive because of technical skills and procedural work [26].

Financial reward and medical student debt may also affect the choice to be a generalist or specialist, though the evidence is mixed. A review suggested higher medical student debt may lead to pursuing higher paying specialties in countries like the USA [29], although other USA [29,30,31] and Australian research [32] contradicts this. 

Demographics may equally overlay choice patterns. Females show differentiated considerations of work-life balance and part-time work options when choosing specialties [11,12,25,26,33]. Females are widely demonstrated to be more likely to work in general practice, which has more flexible work options [33]. Males of older age at medical school graduation may also choose general practice to fit with the rest of their lives [32]. Apart from gender, other factors may ‘prime the pump’ for choosing to be a generalist or specialist, such as ethnic, family and community background as well as personal experiences, but these are under-researched. 

There is minimal research specifically dichotomised to generalist or specialist choice, which accounts for the temporal dimensions impacting choice-making. Only one longitudinal study in the UK suggests general practice interest may increase over time following graduation (18% to 33%), 81% noting this related to achieving particular work conditions and 44% to fit domestic circumstances [33]. Otherwise, the decision-making process regardless of specialty is known to be multi-staged [20] and emergent [22]. 

In summary, complex dynamic patterns are likely to underpin specialty choices but there is minimal theory about how the choice to be a generalist or specialist doctor occurs which accounts for doctor’s characteristics and their experiences over time. We aimed to develop theory about what works for whom, when and in what contexts, to yield choice to become a generalist or specialist doctor. 

## 2. Materials and Methods 

### 2.1. Design

We used a realist evaluation method guided by the RAMESES II standards because our question was realist in nature and realist evaluation is applicable for evaluating complex issues [34]. We aimed to explore how context (C) (the backdrop of the doctor’s personal characteristics and experiences over time) would trigger mechanisms (M) (the things that enable or the generative force) to yield uptake of a generalist or other specialist medical career (O) [35]). The terms used in realist methods are outlined in Box 1. Realist evaluation aims to test initially hypothesised theory and develop and refine new theory about how programs achieve results, frequently expressed as C + M = O configurations (CMO). As such, the outcome of a realist evaluation is theory, depicted by one or many CMO configurations. The main author (BOS) had completed formal realist methods training and BOS and MM had previously applied the method to a program evaluation.

This study had ethical approval from The University of Queensland ethics committee 2012001171.

Box 1Definition of terms used in realist evaluations [35,36].Context—pertains to the backdrop of conditions connected to triggering generative forces (mechanism) that modify behaviour towards the outcome. These may include conditions that change over time, such as funding, trust, experience, locations.Mechanisms—are considered the ‘triggers’ or generative forces that lead to outcomes if they are ‘activated’ in the right conditions. It may denote cognitive or emotional reasoning of the various actors at work, challenges or successes or may be synonymous with the program’s strategies such as responding to an incentive.Outcomes—are intended or unintended resulting from the interplay of context and mechanisms and can be proximal, intermediate or final.Context-mechanism-outcome (CMO) configurations—is a heuristic used to generate causative explanations pertaining to the data. This process draws out and reflects on the relationship of context, mechanism and outcome of interest in a particular program being evaluated. A CMO configuration may pertain to either the whole program or only certain aspects. Configuring CMO patterns is the basis for generating and/or refining theory that is the product of a realist evaluation.

### 2.2. The Environment for our Evaluation

Our study was based in Australia which is experiencing a shortage of generalists and rural doctors related to developing a new National Medical Workforce Strategy which this evaluation can inform [37]. After completing university-based medical training, which is of 4–6 years’ duration (noting Australia has a mix of under and postgraduate medical degree options), doctors work independently in hospitals as pre-registrars for a minimum of 2 years. Around this time, they are eligible to start applying/commence vocational training (spanning 3–6 years), which involves entering a competitive process for selection into one of a number of individually governed medical colleges (equivalent to ‘residency’ in many countries). 

### 2.3. Initial Program Theory

Realist methods require that researchers have an initial program theory, which can be tested during the realist evaluation process. This involves broadly hypothesising the potential causal patterns at play for producing generalist or specialist doctors [35]. We applied reciprocal determinism as part of social cognitive theory to our evaluation question. This theory was set out by psychologist Bandura in 1978 [38]. It notes that a person’s behaviours both influence and are influenced by personal factors like cognition and the social environment such as observing other doctors. Further, the impact on behaviour may be conditioned from what is experienced/observed and the consequences of this, such as negative feedback or low financial reward. This theory aligns with the background literature about specialty choice, showing it is complex and dynamic, impacted by intrinsic and extrinsic drivers [26], an interplay of influences and mediating factors [15,28].

Moreover, that choosing a specialty involves a complex cognitive process undertaken within a personal, social and professional context particular to each individual [28,32] and over different stages [20]. The methods for exploring this further, across two phases, were chosen to firstly allow for in-depth analysis of empirical data from contemporary early career doctors about their career decisions (interviews) (phase 1: developing theory). Secondly, and broader perspectives beyond the context of the individual were collected, by checking phase 1′s findings with a wider sample of medical experts involved in this field, along with exploring other literature (phase 2: refining theory).

#### 2.3.1. Phase 1: Developing Theory

To develop theory relevant to the research question, in 2019, we drew on a purposeful sample of 82 doctors who graduated from the University of Queensland (one of Australia’s largest medical courses) for whom we had Email contact details. We aimed to recruit graduates between their 1st and 17th postgraduate year of work (as this is a broad period of those both entering and recently experiencing specialty training, thus capturing specialty choice decisions across diverse pathways/fields), covering a mix of genders, work locations and generalist/specialist fields to gain a breadth of perspectives of relevance to our research question. 

A semi-structured interview schedule was developed and piloted by the research team of mixed qualitative and quantitative experience and explored “*the nature of medical career decision making”* including reflections (current or recent past) about specialty choice (Table 1). Participants were blinded to the research question to encourage free reflection from the perspective of their own experiences.

Interviews of up to 40 minutes’ duration were done using video and phone-meetings, by two qualitative-PhD-trained female interviewers who had no prior relationship with participants (TG and PM). Participants were not paid. Prompts (Table 1) were used to expand and deepen understanding of issues for full description [39]. Post-interviews, the researchers recorded reflective notes and discussed emerging themes with the wider research team for sense-making and informing hidden areas for further exploration [39]. Data collection ceased once saturation was reached. Interviews were recorded, transcribed verbatim and de-identified using a unique identifier. 

The full de-identified transcripts were read by the whole research team. For a breadth of interpretation, the research team included academics with experience as clinicians (BOS, PM, both non-medical), policy/program staff (BOS, PM) and mixed methods medical workforce research (all). This allowed analysis to draw on different theoretical interpretations of the data (triangulation) to reduce subjective bias [39] and be self-reflexive with respect to predilections or opinions [40,41]. 

The researchers highlighted and sorted CMO configurations from transcripts, building on and expanding the original program theory. These configurations were discussed at multiple meetings (iterative process), where reflective notes were recorded and shared with the team to aid depth of analysis. Thereafter, full transcripts and extracted text were re-reviewed by all authors, to check for any deviations and consider consistent CMO configurations underlying an holistic theory [39]. This process enabled internal corroboration or disconfirmation [42,43] until the research team reached consensus about a coherent phase 1 program theory. 

To aid interpretation, transcripts and extracted text included notation of participant characteristics and the outcome: generalist or specialist choice (Table 2). 

#### 2.3.2. Phase 2: Refining Theory 

Inherent to the realist evaluation method, we sought to check the validity of our phase 1 theory and refine it [34]. To do this, a table of CMO configurations from phase 1 (our first stage of program theory) was sent by Email to other medical generalists and specialist experts from Australia, known for leading medical education and/or publishing in the field of medical workforce education/training. They were purposefully selected for a mix of gender, career stages, medical school of origin and Australian states. Those choosing to respond participated in an informal phone conversation about the theory, approximately one week later, (led by BOS), where the theory was explained and participants were asked to use their own experience/observations to reflect on potential refinements and missing elements. Where new or refined CMO configurations were proposed, they were explored for confirmation with further participants and considered with reference to the existing literature. Final patterns were validated or disconfirmed by in-depth discussion with the research team.

## 3. Results

In phase 1, 32 postgraduate doctors participated, including 50% females and 38% of generalist (11 general practice and 1 public health) and 63% specialist choice (anaesthetics, ophthalmology, surgery, physician, radiology, psychiatry, dermatology) (Table 3). 

In phase 2, all 17 contacted doctors responded including graduates of various Australian medical courses, including 30% who were female. Eight were generalists (seven general practice and one public health) and nine specialists (psychiatry, urology, emergency medicine, anaesthetics and three physicians and two from obstetrics and gynaecology).

Phase 1 identified theory consisting of six CMO configurations depicting six mechanisms that stimulated generalist or specialist career choice. These configurations included three mechanisms of an environmental nature: a conversion; ruling things in or out and; validation and support. Two were of a professional nature: suits desired clinical practice and; fits personality and skills. One was of a non-professional nature: work-life balance and personal sustainability. Phase 2 confirmed this theory (each of the six CMO configurations) and identified two additional CMO configurations that should be added. One was of a professional nature: status and reward and; another of a non-professional nature: suits desired economic and social position. The final refined theory consisted of eight CMO configurations, of which the mechanisms are summarised in Figure 1. The full CMO configurations underpinning the consolidated theory are summarised in Table 4 and described below, by mechanism. 

### 3.1. Environmental

#### 3.1.1. A Conversion

Key focused clinical experiences during medical school were pivotal for choosing to be a specialist particularly if these were reinforced by further exposures in the area of interest:


*I was a medical student…. I visited a surgeon… who ended up doing the most comprehensive face transplant in history… after that… I did a student elective in [major city]—plastic surgery—that was quite good, and then I got into the nitty gritty of trying to be a Plastic Surgery Service Registrar.*
(FM4_Male_Spec)

Some were also converted to specialist fields from a sense of belonging/comradery within a hospital Department: 


*I just clicked with that department. I really enjoyed the people I worked with. I enjoyed the nature of the work, so that’s how I chose anaesthetics.*
(TR1_Fem_Spec)

For generalist choice, early experiences of connecting to a community and rural area were transformative, if reinforced:


*I did a rural health placement here [regional centre] as a student… I wasn’t really interested in GP probably still at that point… but I was really interested in Aboriginal health… I decided to apply for internship up here… then when I was a Resident… I did a PGPPP [general practice rotation] in [remote area]… in a homeland service… which was just incredible.*
(FR5_Fem_Gen)

Phase 2 confirmed this pattern of decision-making was valid and identified that generalist conversions could also be stimulated by contact with exemplary generalist doctors [15,24,43].

#### 3.1.2. Ruling Things in or Out

Choosing a specialist career involved evaluating a range of mostly postgraduate clinical experience for what was enjoyable and ruling things out.

[as a junior doctor]*… it’s just been solidified over time as I’ve done different rotations. And you rule out certain specialties.*(TM2_Male_Spec)

Comparatively, generalists had a degree of difficulty with choosing one area and progressively ruled things in:

[as a junior doctor] *I had trouble choosing one specific specialty… I* [hoped I] *could have that opportunity to practice some primary health, some hospital health in emergency on the wards as well as some anaesthetics and giving me that wide breadth.*(TM1_Male_Gen)

Phase 2 confirmed this pattern of decision-making and added that a generalist choice was a way for the things that doctors ‘ruled in’ to be aggregated under a single role, with sufficient training [44].

#### 3.1.3. Validation and Support

Receiving feedback and endorsement of focused skills, including references from a specialist, was related to choosing to become a specialist. This occurred at a stage when they were impressionable and open to new experiences. 


*I think the primary motivating factor for psychiatry… was driven partly by what I perceive to be reasonable success and good feedback when I worked in a junior stage. I think I was quite impressionable and so, I was quick to jump…*
(TR3_Male_Spec)

For generalists, validation and support came from professional role models (often supervisors) who invested in a personal connection, demonstrating lifestyle and continuity medicine as early as medical school:

[When medical student]*… I was nursed along and shown what the joys of general practice and long-term care in a community was like.*(FR1_Male_Gen)

[When medical student]*… individuals who were prepared to take me into their personal and family lives, and not just at clinic… as a person, in my early 20s, that had a big impact on my ideas about the world.*(FR6_Fem_Gen)

Phase 2 confirmed this pattern of decision-making.

### 3.2. Professional

#### 3.2.1. Suits Desired Clinical Practice

Choosing to be a specialist also occurred when doctors evaluated the suitability of the components of clinical practice against professional expectations like achieving intellectual stimulation, doing procedural work and working in acute hospital care. For doctors of fixed specialty ideation at medical school entry (who knew exactly what sort of specialist they wanted to be), experiencing their preferred specialty reinforced their orientation to that particular specialist field.


*I always loved doing critical care, I was always interested in looking after sick patients. I always wanted to work in a hospital environment. That’s just how I felt about it….*
(FM1_Fem_Spec)

For doctors with malleable career ideation, postgraduate experiences aided an attraction to a particular specialist area:


*I became interested in anaesthetics when I was in my intern year… I guess I really enjoy the very procedural nature of anaesthetics*
(TR1_Fem_Spec)

Choice to be a generalist was fashioned by evaluating clinical practice against professional expectations of using a breadth of skills, being involved in holistic and longitudinal patient care improving population health. This mainly occurred in the postgraduate stage.

[As a junior doctor]*… I can do whole of life care and get in earlier and be the first point of contact rather than just see people when they get to hospital.*(FR6_Fem_Gen)

For some, the desire to work in a generalist role to make an upstream difference emanated from getting burnt out by acute hospital healthcare:

[As a junior doctor]*… I was burnt out from the hospital—you see all the sort of pointy end of things there.*(FR5_Fem_Gen)

Phase 2 confirmed this pattern of decision-making and expanded that choice to be a specialist was also related to desire to work in teams [28] whereas choosing to be a generalist was related to seeking more autonomous decision-making [16,32,43]. 

#### 3.2.2. Fit Personality, Skills and Norms

Doctors choosing to be a specialist discussed being drawn to a field that they perceived fit their attributes, whether these were technical (knowledge of anatomy) or soft skills (communication). 

[when a junior doctor] *I chose oncology… I guess my communication skills are probably my strongest point and oncology is a specialty where it’s based around communication.*(FR8_Male_Spec)

Few choosing to be a generalist noted particular personality or skills that drew them to this, except being comfortable with uncertainty. Phase 2 confirmed this pattern of decision-making and added that along with personality and skills, doctors also evaluated the fit of particular fields to desired professional norms. Those choosing to be a specialist were more likely to desire to align with professional norms [28] whereas generalists, to challenge these included integrating traditional siloes of medical care under one practice model (see *Collingrove Agreement*) [45]. Further, extending on their ‘comfort with uncertainty’, doctors choosing to be a generalist have attributes of enjoying problem-solving, innovation and change [43,46]. 

#### 3.2.3. Status and Reward

Phase 2 identified a new pattern of decision-making about status and reward, which was validated through further testing and relating to the literature. This occurred in medical school and was reinforced over time. Doctors oriented to specialist choice were sensitised to the inferiority of generalists after hearing from other (hospital) doctors that generalist skills were less, commencing in medical school and reinforced over time [15,24,32]. Those with a desire to be known for doing one thing well (professional status), and to maintain income in a tightly controlled professional network and market, were stimulated to choose to choose to be a specialist [16,28]. People with healthcare power are known to be more likely to act to increase this power including by talking others down, negotiating and using coercion, to maintain this [47,48].

Status and reward influenced choice to be a generalist where doctors observed generalists with excellent skills, recognised by a professional title and well remunerated and supported to use all their skills (capacity to maintain income in a broader market and sustainable rosters and back up supports). This included observing that being able to do many things well achieved status in the community, and made a doctor useful [49]. Recognition methods necessarily have to handle the competing identities of doctors working under the generalist banner (rural and non-rural generalist practitioners is one distinction) and reconcile historical and aspirational conceptualisations of their roles [50]. 

### 3.3. Non-Professional

#### 3.3.1. Work-Life Balance and Personal Sustainability

Mainly at the postgraduate stage, female and male doctors chose to be a specialist in a particular field, to fulfil expectations for controlled working hours. Males mentioned this in relation to firstly, lifestyle and secondly, being older when they completed medicine and wanting to set up practice faster.

[with partner and children] *Oncology … was a specialty that appealed to me…for a bit of a lifestyle—not a lot of after-hours.*(FR8_Male_Spec)

[psychiatry] *I was very well supported in paediatrics as a PHO, but I looked at how long the training programme was at my age and what I’d have to learn and I, despite their assistance, I didn’t go that way.*(FM5_Male_Spec)

Females did this if they had a partner and were planning children, desiring a sustainable role around personal goals.

[partner planning children, anaesthetics]*… a career that I can spend time with my children when I have them and all that, and spend time with my partner… you don’t have inpatients, you don’t have longitudinal care… it doesn’t drain you…*. (JM1_Fem_Spec)

Females chose to be a generalist for work flexibility and part-time hours:


*… my own health and then also the birth of my son, yeah just helped to cement my desire for a more flexible part-time approach to clinical work.*
(TR3_Fem_Gen)

Males chose to be a generalist if they wanted shorter times to access and greater ease to complete training thus commencing independent practice sooner. One participant who was older had considered ‘Emergency medicine’ but saw ‘tough training’ and chose to be generalist for flexibility and part-time options.

[GP]*… allowed much more flexibility in the training and taking part-time work, for example, which any of the other specialties didn’t allow.*(FR4_Male_Gen)

Phase 2 confirmed these decision-making patterns, including the nuanced differences by gender. Other literature identifies that female doctors favour sustainable careers [49,50,51] and that male doctors choose careers that allow for lifestyle interests, not restricted to having/raising children [16].

#### 3.3.2. Suits Desired Economic and Social Position

Phase 2 identified a new pattern of decision-making about suiting desired economic and social position, which was validated by further testing and in relation to the literature. Doctors chose to be a specialist based on observing the positive socio-economic benefits of various fields. A perception of improved economic and social position was forged by early experience within medical families, at medical schools and reinforced over time, when doctors socialised and worked together [28]. Those with a desire to improve or uphold their socio-economic position and achieve financial security through a medical career, were attracted to specialist roles which pay more than generalist roles [52]. This desire was potentially reinforced by the level of expected rewards for the cost and effort related to training as a doctor [53] and the working hours involved in the role [54].

For doctors choosing to be a generalist, their desired economic and social position was considered in relation to broader socio-cultural values that were wider than gains to be made within the profession [55]. This could include prioritising and complementing other aspects of their socio-cultural identity formed by the values they held for family and within wider society, beyond a professional identity [24]. Other literature confirmed that generalist doctors are more motivated by benevolence, than money and power [56], suggesting that for generalists, social and cultural interests may be stronger than economic ones. 

## 4. Discussion

This is the first known study to develop theory about choosing a generalist or specialist medical career. The decision-making patterns revolved around eight mechanisms of environmental, professional and non-professional domains. These may contribute in proximal, intermediate and final ways [35], to achieving a generalist or specialist doctor, depending on the doctor’s characteristics including their attributes, values and desires and how these intersect with their exposures over time. 

The final theory reinforces, with some degree of nuance, elements of the original hypothesis about how choice is made, through the theory of reciprocal determinism. This includes depicting that personal cognitive, social/environmental components and conditioning plays a strong role in generalist or specialist choice [38]. Various CMO configurations have the potential to work in synchrony and nudge towards a tipping point of choice to be a generalist or specialist doctor, particularly where these may intersect and build momentum over time. No one CMO configuration within the theory is considered causal, but together these configurations contribute to the emergence of generalist or specialist choice.

Some triggers were stronger for some doctors than others. But our findings provide an understanding of a full range of ways that choice-making can be affected. This includes the context of the doctor and timing by which choice is triggered, whereby our findings have the potential to holistically inform education, training and workforce strategies for better uptake of generalist doctors and the distribution of rural doctors [7,37,57]. 

Although we present this theory as driving the outcome (positive direction), it can also produce negative outcomes, if patterns of generalist decision-making are suppressed, or insufficient triggers are mounted. Thus, the theory may have greatest utility if used to design holistic policies and programs that promote multiple pro-generalist decision patterns and dampen many of the pro-specialist ones. 

Our initial theory was strengthened by drawing on empirical evidence from recent graduates (all of whom at chosen specialty) across a spread of specialties, genders and locations. By then gaining further input from experts spanning different medical schools, career stages and disciplines, enabled the findings and perspectives of individuals to be refined and expanded, supporting greater generalisability of the final theory. This builds on existing research showing specialty choice is multi-level [26] and multi-staged [20], by uniquely depicting the timing of various program, social-economic and cultural normative influences on driving to a generalist or specialist outcome. 

The findings identify that exposures for choosing a generalist career such as connecting ‘*to a community*’ and ‘*role models*’, may require recurrent investment (including in medical program design) and be strong and frequent enough to override stimuli leading to specialist choice. This includes reducing the potential that some pro-specialist triggers could fire including doctors being converted by ‘*key focused clinical experiences*’ with specialist departments in hospitals. Other research shows the value of community general practice placements for pre-registrar doctors during internship (additional knowledge and skills) [58]. Planned and regular rotations to non-hospital settings, including in rural areas, with exemplary skilled generalists, who showcase innovative practice, ‘*problem-solving*’ and procedural aspects of their work have the potential to stimulate generalist career interest. Students and junior doctors may also be inspired if they observe the status of generalist doctors in the community, respected for their confidence and competence in a range of situations. This needs to be powerful enough to override potential professional derision of generalists by specialists who are seeking to maintain professional power and market control [15,47]. 

Our findings also depict that choosing to be a generalist also relies on getting ‘*enough experience*’ of different forms of clinical medicine to ‘*rule things in*’. This differs from the perception that generalist doctors take this path because they aren’t sure about what to do (path of least resistance). On the contrary, generalists are likely to choose this deliberately ‘*ruling in*’ a package of skills areas that form a complementary clinical practice model that is remunerated, recognised, sustainable and allows them to focus on upstream health improvement [59]. Conceptualising viable generalist practice models may take longer for junior doctors than understanding work in more homogenous areas like hospital specialist fields that have a clear professional identity. This may underpin the need for a longer pathway and more deliberate exposure to potential models in areas of interest, to stimulate a generalist choice. 

Several elements of theory relate to contemporary challenges. In many countries, more doctors are emerging from postgraduate medical degrees, having incurred more time and cost to achieve two degrees to qualify as a doctor than those from undergraduate systems. Our theory might suggest that older graduates may be more likely to drive towards choosing particular specialty fields or generalist practice, based on two factors: interest in a rapid transition to independent practice (shorter training times and relative ease of training) and to manage work-life balance (leisure, children or other constraints like illness). The tipping point for this group to nominate to a specialty field is that some of these fields enable controlled hours (noted from our research, as psychiatry, anaesthetics and oncology). For this reason, a generalist choice cannot rely on controlled working hours and flexible conditions alone to attract doctors. Instead it requires multi-level strategies including emphasising the gains of organised training pathways to rapid independent practice and promoting of the gains for choosing a generalist career, such as community recognition for ‘*doing many things well*’. This could be strongly promoted as part of messaging within national campaigns. 

Although specialists may claim legitimacy based on their lengthy professional training, expert status and certainty in one area, it may be important to counter this with evidence of generalist competence [48], trust and credibility [60] and the reward generalists may experience from contributing to social (not just professional) goals. This may be important for breaking down the assumed professional hierarchies and levels of reward enabled in specialist roles [28]. Further a structural issue to address, is reducing the gap in earnings between specialists and generalists [52]. 

As hours of medical work are trending down (average fall of 3.4 h per week 1999–2009 in Australia) [61], advertising generalist work through access to shorter training time frames, flexible and part-time work tailored to trainee needs (including gender-specific flexibility and maternity leave) and sustainable practice models (minimising burnout) continues to be relevant. This issue is increasingly pressing as females (wanting to build careers around children) are making up the bulk of emerging medical school graduates in many countries [62,63,64]. 

Finally, our findings also suggest that generalists may be achieved by enrolling more students into medicine who have wider values and social interests based on family, culture and community, as the basis of their identity (status), over would-be-doctors motivated by professional identity and socio-economic gain [28]. Given that values and expectations are established within a socio-cultural context of family, ethnicity, religion and community, it may be relevant to consider these as important covariates that can affect generalist workforce outcomes.

Our study has limitations. Although we used a 2-phase process to build and refine our theory, it is possible that some elements of theory were missed. This is unlikely given that the cross-university cross-career stage experts in phase 2 largely supported the phase 1 theory, expanding only to two new patterns of decision-making that were cross-validated. Relying on phase 1 interviews across a broad single university early career cohort means there is some potential for sample and recall bias. However, participants were working independently of the university when interviewed and easily recalled their career choice process, whether generalist or specialist and, being blinded to the research question, provided genuine reflections. 

The theory we propose is based on medicine in Australia and needs to be refined and validated for other disciplines, countries or career stages. This is particularly because in some countries like America and Canada, the timing of generalist or specialist career choice may occur earlier as part of filling particular pre-set generalist or specialist programs in medical schools that articulate with resident programs, which does not occur in Australia. 

In our theory socio-cultural and familial influences mostly featured in relation to affecting pre-set personality, norms and skill as well as the desire for social and economic position relative to other values, but their role and timing of socio-cultural and familial influences may vary in different training sub-systems, countries and cultures. As it was based on a dichotomous outcome, out theory may also require further differentiation for choosing specific specialties and sub-specialties of medical work, including exploring whether this theory applies to further differentiating choice to be a more general (e.g., general surgeon, or more focused sub-specialist e.g., paediatric cardiologist.

## 5. Conclusions

Our study developed new theory about the dynamics of choosing to be a generalist or specialist doctors. Within three domains: environmental, professional and non-professional, we found eight clear mechanisms linked with the patterns of decision-making to yield a generalist or specialist outcome. These represent multi-level triggers which are turned on by various exposures, relative to doctor’s characteristics, at different times, to determine generalist or specialist choice. The findings provide an avenue for tailoring medical education and postgraduate work programs, as well as selecting and mentoring students and junior doctors with particular attributes, norms, values and professional orientations, to increase generalist uptake. 

## Figures and Tables

**Figure 1 ijerph-17-08566-f001:**
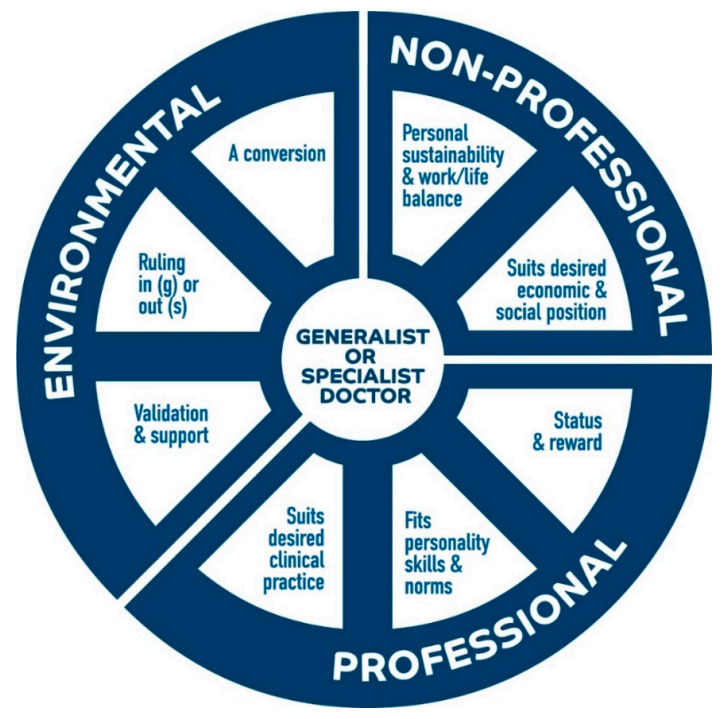
Mechanisms to produce a generalist or specialist doctor. For the mechanism ruling in or out.

**Table 1 ijerph-17-08566-t001:** Interview guide used in phase 1.

Question	Specific Prompts	General Prompts
Could we start by you telling me a little about yourself and your career as a doctor?	Things like your current practice location, area of medicine, stage of medical career, and where you did each stage of your medical training?	Could you please expand on that? That is very interesting, could you tell me more? Really, what was that like? Reflecting on that time in X, could you give me a bit more detail about X experience?
What are the major factors that have influenced your medical career journey to date?	Identify factors that influenced participant’s career decision, current practice location; area of clinical practice; amount of time devoted to clinical medicine; decision-making in the context of family situations, partner employment, incentives, professional support	
What were the important time points when things happened that determined the current shape of your medical career?		
What made these time points important?		
What happened at those times and how did they affect your career trajectory?		
How much control have you had over how your medical career has turned out?	Things like; going to medical school, internship location, vocational training, geographical location of current clinical practice	
What are the factors that influenced (gave you more or restricted) that control?		
How easy (or realistic) is it to change where you practice (geographically); and also your field of medical practice?	How flexible is a medical career; and does it vary at different times in one’s life? Does it vary by area of medical practice? By where you live (city/country)?	
Have you considered changing where you practice or your field of medical practice?		
Have you had to move from where you were living to pursue a training opportunity, or to meet clinical/professional college requirements?		
Did you later return to where you were?		
Have you had breaks in practice?		
Can you tell me the reasons for those breaks?		
What would have made your medical career progression better informed?		
What (else) would have improved the way your medical career has progressed?		
Before I turn off the recording device, is there anything else you would like to comment on?		

**Table 2 ijerph-17-08566-t002:** Definition of notation used to depict participants in the text of phase 1 interviews ^a^.

Notation	Definition
J, T or F	junior doctor, trainee or fellow (defined in Table 3)
R or M	working rurally or metropolitan
Male or Fem	male or female
Gen or Spec	Generalist (general practice or public health) or specialist (all others) based on self-reported interest/uptake of a chosen postgraduate field of medicine

^a^ All participants interviewed had decided on, commenced or recently completed a specialty field allowing the outcome to be measured. Rural work location was determined using official Modified Monash Model levels 2–7, which is the standard definition used by the Australian government for health policy [44].

**Table 3 ijerph-17-08566-t003:** Summary of phase 1 participants (*n* = 32) ^a^.

Characteristics	*n* (%)
**Sex**	
Females	16 (50)
Males	16 (50)
**Training stage**	
Junior—yet to start vocational training as registrar (typically PGY 1–5)	8 (25)
Trainee—currently enrolled in specialty training (registrar) (typically PGY 3–10).	10 (31)
Fellow—completed specialty (registrar) training (typically PGY 6–17)	14 (44)
**Working rurally**	
Yes	15 (47)
No	17 (53)
**Rural background**	
Yes	8 (25)
No	24 (75)
**Specialty focus**	
Generalist	12 (38)
Specialist	20 (63)

^a^ Rural work location was determined using official Modified Monash Model levels 2–7, which is the standard definition used by the Australian government for health policy [44]. All participants interviewed had decided on, commenced or recently completed a specialty field allowing the outcome to be measured. ‘Generalist’ includes doctors interested, training or fellowed in general practice or public health physician. ‘Specialist’ included doctors interested, training or fellowed in focused fields –interviewees covering anaesthetics, ophthalmology, surgery, physician, radiology, psychiatry, dermatology.

**Table 4 ijerph-17-08566-t004:** Full theory about exposures (C) for doctors at different stages of training (C) triggering choice (M) to be a generalist or specialist doctor (O) **^a^**.

Outcome	Trigger for Choice (Mechanisms)	Doctor’s Characteristics/Timing of Exposure (Context)	Doctor’s Exposure (Context)
Specialisation choice (S or G)	ENVIRONMENTAL		
	A conversion	(S/G) Medical school and reinforced over time	(S) A key focused clinical experience or clicking with a Department or specialist clinician (G) Connecting to a community and/or rural areas and exemplary generalist clinicians
	Ruling things in (G) or out (S)	(S/G) Mostly postgraduate	(S) Experiencing a range of areas of clinical medicine (G) Experiencing a range of areas of clinical medicine and seeing how these can be linked into generalist practice, with sufficient training
	Validation and support	(S) Early postgraduate when impressionable (G) Medical school and early postgraduate when impressionable	(S) Getting reinforcing feedback from senior clinician/s, focused clinical skills and endorsement/references for job/training applications (G) Connecting with role models who invest in a personal relationship, demonstrating lifestyle and continuity medicine
	PROFESSIONAL		
	Suits desired clinical practice	(S) Mostly postgraduate if do not have a fixed specialty ideation (S) Medical school if have a fixed specialty ideation (G) Mostly postgraduate, burnt out from hospital work	(S) Being intellectually stimulated, enjoying procedural work and working in acute hospital care and comfortable with working in teams (G) Enjoying skills breadth (including procedural and intellectually challenging work), complexity of the ‘whole person’ continuity of care, working independently and making an upstream impact to population health
	Fits personality, skills and norms	(S) Before medicine, medical school and postgraduate (G) Mostly postgraduate	(S) Having particular attributes—technical or soft skills and desire to align with social and professional norms (G) Comfortable with uncertainty and enjoy problem-solving, innovation, change and challenging social and professional norms
	Status and reward	(S) Medical school and reinforced over time, desire to optimise professional power and maintain income through market control (G) Medical school and reinforced over time, desire to be useful and maintain income within broader market	(S) Being sensitised that G have inferior skills and observing benefits of being known in tight professional network for doing a key skill well (G) Observing G with excellent skills (recognised by professional title) and remunerated/supported for the range of their skills, working in sustainable models (enough clinical back up), plus benefits of being known in community for doing many things well.
	NON-PROFESSIONAL		
	Work-life balance and personal sustainability	(S–M) Mostly postgraduate, partner and older when completed medical school (S–F) Mostly postgraduate, have partner, planning/have children (G–M) Mostly postgraduate, partner and older when completed medical school (G–F) Mostly postgraduate, partner, planning/have children and/or other personal constraints	(S–M) Observing specialty options with controlled working hours and feasible to complete (length, difficulty) (S–F) Observing specialty options with controlled working hours and less job creep into personal life (G–M) Observing shorter times to access/complete training and flexible and part-time work options (G–F) Observing flexible and part-time work options
	Suits desired economic and social position	(S) Medical school and reinforced over time, desire to gain or uphold social status and financial security relative to familial and social expectations, cost/effort of training and potential remuneration for the working hours involved (G) Medical school and reinforced over time, desire to uphold broader socio-cultural values including important non-professional roles	(S/G) Observe benefits of socio-economic position

^a^ Rural work location was determined using official Modified Monash Model levels 2–7 of the Australian government [44]. G refers to ‘Generalist’ and includes doctors interested, training or fellowed in general practice or as public health physicians. S refers to ‘Specialist’ and includes doctors interested, training or fellowed in focused fields –interviewees covering anaesthetics, ophthalmology, surgery, physician, radiology, psychiatry, oncology, dermatology.

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
