# Peer review of "A Realist Evaluation of Theory about Triggers for Doctors Choosing a Generalist or Specialist Medical Career"

_ijerph, 2020, doi:10.3390/ijerph17228566_

Round 1

Reviewer 1 Report

Overall, this is an interesting and timely article on choosing generalist or specialist as a physician in Australia. 

The development of the theoretical framework is exemplary.  Thank you. 

While I am fairly certain the theoretical framework is relevant for other countries, it will be interesting to see if future work in other settings bears this out. 

I would encourage the authors to:

1) provide a bit more background on medical education in Australia (I had to go look up the educational system online to understand what the authors were talking about in terms of "basic medical training" and working as "registrars") and

2) discuss how this may vary in other medical education systems a bit more fully in the discussion section.  

Otherwise, this is a very good article on a timely topic.

Author Response

Reviewer 1

Response

Overall, this is an interesting and timely article on choosing generalist or specialist as a physician in Australia. 

The development of the theoretical framework is exemplary.  Thank you. 

Thank you

While I am fairly certain the theoretical framework is relevant for other countries, it will be interesting to see if future work in other settings bears this out. 

We agree and note our existing comment in the limitations that "the theory, based on medicine in Australia, would need to be refined and validated for other disciplines, countries or career stages."

I would encourage the authors to:

1) provide a bit more background on medical education in Australia (I had to go look up the educational system online to understand what the authors were talking about in terms of "basic medical training" and working as "registrars") and

We added more details to this section, under Methods, as underlined here:

"The environment for our evaluation"

“…Our study was based in Australia which is experiencing a shortage of generalists and rural doctors related to developing a new National Medical Workforce Strategy which this evaluation can inform (39). After completing university-based medical training, which is of 4-6 years duration (noting Australia has a mix of under and postgraduate medical degree options), doctors work independently in hospitals as pre-registrars for a minimum of 2 years. Around this time they are eligible to start applying/commence vocational training (spanning 3-6 years), which involves entering a competitive process for selection into one of a number of individually governed medical colleges (equivalent to ‘residency’ in many countries).”

2) discuss how this may vary in other medical education systems a bit more fully in the discussion section.  

It is not possible to comment on all possible medical education systems; however, we have particularly added commentary in the Discussion, within the limitations section noting that differences may emerge in a North American system where the timing of residency training may be closely connected to enrolment in medical school training.

“… in some countries like America and Canada, the timing of generalist or specialist career choice may occur earlier as part of filling particular pre-set generalist or specialist programs in medical schools that articulate with resident programs, which does not occur in Australia.”

Otherwise, this is a very good article on a timely topic.

Thank you

Reviewer 2 Report

This paper is on an important topic both nationally in Australia, and Internationally, where ever medicine is an elite career.  The downstream implications are that it could drive selection criteria to suit both choice and longevity in a preferred generalist/specialist career path, and to develop junior doctors in particular directions – though the ethics of doing this would be rather questionable!

It is true that the space on career decisions is largely a-theoretical, although there is actually a large body of literature on specific correlates of different disciplines, which is not mentioned or covered.  Additional factors could have been suggested with respect to “priming the pump” (eg medical students’ ethnic background, personal experiences, media exposure etc) – each of which could be independent of medical school, but likely to be influential, so the suggestion that triggers do not start until medical school is probably a little “trigger happy”!

I wonder whether the proposed model actually presents a theoretical understanding, or is it more of a descriptive model?  It takes a clear realist approach methodologically, but in this, there is no overarching theory being critiqued, nor one being developed, other than that there are “triggers” which predispose a medical student/doctor to make particular generalist / specialist decisions.  It seemed a bit presumptuous to label what were called “triggers” as theoretical mechanisms.  They were clearly themes with clear associated quotes.   But to speak of a trigger presupposes a mechanism - a “trigger” is a causal concept – which is not defined in the paper (eg it it a feeling? a thought? a social entity?  Under what circumstances is a trigger effective vs ineffective?).  There was also no real consideration of how these parts/ “triggers” work in synchrony or otherwise, how they are how affected by age, stage, race, context?  

It would have been interesting for the authors to comment on the effects of background, culture and gender, as an account that could accurately be called a “mechanism” would explain why these factors are important (as they are in the specialist/generalist decision), how they work and by what synergies they orchestrate behaviour into particular trajectories. 

In this regard, for the Australian context, with growing levels of subspecialist practice, the binary division between generalism and “specialism” is a bit coarsely drawn and probably needs to be further defined  (eg an Adult Medicine physician could be just that – and so could probably be called “a generalist”, or by the same token, they could be an adult interventional cardiologist with a subspecialty interest in imaging and treating cardiac valve issues ie “a sub-specialist”).  Could the ideas in this paper give a reasonable account of whether  general specialists more similar to "generalists" or to "specialists"?

Since the model is not really complex enough to explain whether the proposed factors exist in a hierarchy or series, or how efficacious the different triggers might be on producing the given choice for generalism or specialism, it would be rather difficult to test.  Why don’t all medical students with rural experience, and strong community involvement, become generalists? In such a small sample, simply observing a shared perspective, as this study clearly does, is not the same as being able to imply causation, which a theory can and must do.

Interesting omission of the large literature on what is known about generalist career correlates, which include female gender, rural orientation, predominance of training in primary care prior to choice, low socioeconomic status, ethnic status, remuneration – some of which are mentioned, but not in a way that could explain the decline of interest in generalism, or a method to turn the tide.  Could the authors try to draw these kinds of causal statements from their work?

Methodologically, the sample represented an interesting choice of PG years – why terminate in PGY17 in particular?  Might the sample in fact be a random one being put forward rather disingenuously as purposive?  It would have been interesting to stratify the sample by age and discipline as overall, generalist status has been steadily declining since 1970s in Australia, save for generalism in certain work contexts.  Another methodological comment is that the specialties chosen were not particularly representative of the specialist workforce in Australia, and included disproportionate numbers of women, primary care and rural practitioners.  Was the sampling stopped at saturation?

Only cursory attention was paid to those entering med school with a career decision already substantively made – i believe there is clear existing evidence from America around the hierarchical order of experience associated with primary care choice that begins before medical school.  It is a little presumptuous to puts the medical school as the origin of the generalist/specialist choice..

Some of the ideas developed in the paper are very valuable including defining generalism as “counting things in” (as opposed to failing or counting things out ) and  by “doing many things well” (again this positive spin is welcomed – and presumably could be tested.)

The authors are to be commended for keeping all the terms they used deliveratily neutral, and non-pejorative throughout. 

The authors may find the recent popular book by David Epstein worthy reading:  Range, why generalists triumph in a specialised world.

Author Response

Reviewer 2

Response

This paper is on an important topic both nationally in Australia, and Internationally, where ever medicine is an elite career.  The downstream implications are that it could drive selection criteria to suit both choice and longevity in a preferred generalist/specialist career path, and to develop junior doctors in particular directions – though the ethics of doing this would be rather questionable!

Thank you for your positive comments.

It is true that the space on career decisions is largely a-theoretical, although there is actually a large body of literature on specific correlates of different disciplines, which is not mentioned or covered.  Additional factors could have been suggested with respect to “priming the pump” (e.g. medical students’ ethnic background, personal experiences, media exposure etc.) – each of which could be independent of medical school, but likely to be influential, so the suggestion that triggers do not start until medical school is probably a little “trigger happy”!

The introduction provides comprehensive background (36 references) as to the existing literature about correlates of different disciplines and choice-making processes. We revised some of the text to include the term ‘correlates’.

“The literature highlights that choosing a specialty is a complex process with a number of identified correlates.”

We added to the paragraph about demographics:

“Apart from gender, other factors may ‘prime the pump’ for choosing to be a generalist or specialist, such as ethnic, family and community background as well as personal experiences, but these are under-researched.”

I wonder whether the proposed model actually presents a theoretical understanding, or is it more of a descriptive model?  It takes a clear realist approach methodologically, but in this, there is no overarching theory being critiqued, nor one being developed, other than that there are “triggers” which predispose a medical student/doctor to make particular generalist / specialist decisions. 

Thank you for commending this is a true reflection of the realist method.

We added detail of the over-arching theory being critiqued in the methods section under ‘initial program theory’.

“We applied reciprocal determinism as part of social cognitive theory to our evaluation question. This theory was set out by psychologist Bandura in 1978 (41). It notes that a person’s behaviours both influences and is influenced by personal factors like cognition and the social environment such as observing other doctors. Further, the impact on behaviour may be conditioned from what is experienced/observed and the consequences of this such as negative feedback or low financial reward. This theory aligns with the background literature about specialty choice, showing it is complex and dynamic, impacted by intrinsic and extrinsic drivers (26), an interplay of influences and mediating factors (15) (30). Moreover, it involves a complex cognitive process undertaken within a personal, social and professional context particular to each individual (27) (30) and over different stages (20). The methods for exploring this further, across 2 phases, were chosen to firstly allow for in-depth analysis of empirical data from contemporary early career doctors about their career decisions (interviews) (phase 1: developing theory). Secondly, broader perspectives beyond the context of the individual were collected, to build on phase 1’s findings with a wider sample of medical experts involved in this field, along with exploring other literature (phase 2: refining theory).

It seemed a bit presumptuous to label what were called “triggers” as theoretical mechanisms.  They were clearly themes with clear associated quotes.   But to speak of a trigger presupposes a mechanism - a “trigger” is a causal concept – which is not defined in the paper (eg it it a feeling? a thought? a social entity?  Under what circumstances is a trigger effective vs ineffective?).  

Under the Methods section “Design” we had already included a clear description of what a trigger is but this may have been missed in the review process:

“(the things that enable or the generative force) to yield uptake of a generalist or other specialist medical career”.

For further clarification to readers less familiar with the realist nomenclature, we have added box 1 to provide definitions of realist terms for easy reference.

We note that critical realism argues that ‘the world is characterized by emergence’ of outcomes based on social and cognitive responses as ‘triggers’ to producing an outcome. The word ‘triggers’ is a common term in realist evaluation, used within the realm of discussing mechanisms, which are now also defined in box 1. The outcome of realist evaluations is theory, depicted by one or many CMO configurations, now outlined in box 1 to better guide the reader.

We also added this to Methods section ‘design’

“As such the outcome of a realist evaluation is theory, depicted by one or many CMO configurations”

There was also no real consideration of how these parts/ “triggers” work in synchrony or otherwise, how they are how affected by age, stage, race, context?  It would have been interesting for the authors to comment on the effects of background, culture and gender, as an account that could accurately be called a “mechanism” would explain why these factors are important (as they are in the specialist/generalist decision), how they work and by what synergies they orchestrate behaviour into particular trajectories. 

In response, we added to the paragraph in the Discussion, the underlined section:

“…This is the first known study to develop theory about choosing a generalist or specialist medical career. The decision-making patterns revolved around eight mechanisms of environmental, professional and non-professional domains. These may contribute in proximal, intermediate and final ways (38), to achieving a generalist or specialist doctor, depending on the doctor’s characteristics including their attributes and desires and how these intersect with their exposures over time. The final theory reinforces, with some degree of nuance, elements of the original hypothesis about how choice is made, through the theory of reciprocal determinism. This includes depicting personal cognitive, social/environmental components and conditioning that occurs, plays a strong role in choice (41). Various CMO configurations have the potential to work in synchrony and nudge towards a tipping point of choice to be a generalist or specialist doctor, particularly where these may intersect and build momentum over time.”

In response to the suggestion to comment on the effect of background, culture and gender there are already a number of expressions of this in table 4 theory including in the section “non-professional factors”.

We also added more to the Discussion as outlined below:

“…Further, our theory suggests that generalists may be achieved by enrolling more students into medicine who have wider values based on family and community, as the basis of their identity (status), than seeking professional identity and socio-economic gain (30). Given that values and expectations are established within a socio-cultural context of family, ethnicity, religion and community, it may be relevant to consider these as important covariates that can affect generalist workforce outcomes.”

For the Australian context, with growing levels of subspecialist practice, the binary division between generalism and “specialism” is a bit coarsely drawn and probably needs to be further defined  (eg an Adult Medicine physician could be just that – and so could probably be called “a generalist”, or by the same token, they could be an adult interventional cardiologist with a subspecialty interest in imaging and treating cardiac valve issues ie “a sub-specialist”).  Could the ideas in this paper give a reasonable account of whether  general specialists more similar to "generalists" or to "specialists"?

We agree with your comment, we added further commentary to the Discussion, limitations section:

“…As it was based on a dichotomous outcome, this theory may require further differentiation for choosing specific specialties and sub-specialties of medical work, including exploring whether this theory applies to further differentiating choice to be a more general specialist (e.g. general surgeon) or more focused sub-specialist (e.g. paediatric cardiologist).”

Since the model is not really complex enough to explain whether the proposed factors exist in a hierarchy or series, or how efficacious the different triggers might be on producing the given choice for generalism or specialism, it would be rather difficult to test.  Why don’t all medical students with rural experience, and strong community involvement, become generalists? In such a small sample, simply observing a shared perspective, as this study clearly does, is not the same as being able to imply causation, which a theory can and must do.

Having a number of interlocking CMO configurations emerging from this realist evaluation, as a theory, is a legitimate outcome of a realist evaluation. Yet we note the interest in trying to intersect these in a hierarchy, our phase 2 testing suggested each was legitimate, some weighted more heavily for some doctors over others as a major or minor trigger. We added this into the Discussion:

“…Whilst each CMO configuration was justified as important, some triggers were stronger for some doctors over others. By our findings provide an understanding of a full range of ways that choice-making can be affected. This includes the context of the doctor and timing by which choice is triggered, whereby our findings have the potential to holistically inform education, training and workforce strategies for better uptake of generalist doctors and distributions of rural doctors (40) (62) (7).”

Instead of seeking to imply ‘causation’, our work follows critical realism which argues that ‘the world is characterized by emergence’ where the causal powers that lie within processes are either ‘activated or remain dormant’ to produce an overall impact. As such, no one CMO configuration is considered causal, but together contributed to the emergence of generalist or specialist choice. To this extent we added:

“No one CMO configuration within the theory is considered causal, but together these configurations contribute to the emergence of generalist or specialist choice.”

To help to position our findings, we have expanded the Discussion to position our findings within the initial program theory of reciprocal determinism theory of behaviour change.

“The final theory reinforces, with some degree of nuance, elements of the original hypothesis about how choice is made, through the theory of reciprocal determinism, including personal cognitive, social/environmental components, and conditioning over time (41).”

Interesting omission of the large literature on what is known about generalist career correlates, which include female gender, rural orientation, predominance of training in primary care prior to choice, low socioeconomic status, ethnic status, remuneration – some of which are mentioned, but not in a way that could explain the decline of interest in generalism, or a method to turn the tide.  Could the authors try to draw these kinds of causal statements from their work?

In the introduction, this is mentioned already:

“Females are widely demonstrated to be more likely to work in general practice, which has more flexible work options (36). Males of older age at medical school graduation may also choose general practice to fit with the rest of their lives (35).”

Also key literature about general practice choice is also outlined:

“General practice is also attractive because of lifestyle, continuity of care, procedural skills and work opportunities (27) (15) (16) (28). Primary care role models and experiences may facilitate uptake (29) (15) (28) (24), although scant studies suggest it has lower professional status compared with focused specialties (16) (15) (24).”

Our findings provide some basis for informing pathways to trigger a generalist choice and mitigate a specialist one, with no one being causal on its own (as noted in response to previous comment). The implications for intervening are discussed in the Discussion section.

Methodologically, the sample represented an interesting choice of PG years – why terminate in PGY17 in particular?  Might the sample in fact be a random one being put forward rather disingenuously as purposive?  It would have been interesting to stratify the sample by age and discipline as overall, generalist status has been steadily declining since 1970s in Australia, save for generalism in certain work contexts.  Another methodological comment is that the specialties chosen were not particularly representative of the specialist workforce in Australia, and included disproportionate numbers of women, primary care and rural practitioners.  Was the sampling stopped at saturation?

In the Methods section, Phase 1: Developing theory, we clarify that we did purposefully aim for up to PGY17 and added why:

“to recruit graduates between their 1st and 17th postgraduate year of work (as this is a broad period of those both entering and recently experiencing specialty training, thus capturing specialty choice decisions across diverse pathways/fields), covering a mix of genders, work locations and generalist/specialist fields to gain a breadth of perspectives of relevance to our research question. “

Only cursory attention was paid to those entering med school with a career decision already substantively made – i believe there is clear existing evidence from America around the hierarchical order of experience associated with primary care choice that begins before medical school.  It is a little presumptuous to puts the medical school as the origin of the generalist/specialist choice..

Adding to our current limitations:

“The theory, based on medicine in Australia, would need to be refined and validated for other disciplines, countries or career stages. This is particularly because in some countries like America and Canada, the timing of generalist or specialist career choice may occur earlier as part of filling particular pre-set generalist or specialist programs in medical schools that articulate with resident programs, which does not occur in Australia. In our theory socio-cultural and familial influences mostly featured in relation to affecting pre-set personality, norms and skill as well as the desire for social and economic position relative to other values, but the role and timing of socio-cultural and familial influences may vary in different training sub-systems, countries and cultures.”

Some of the ideas developed in the paper are very valuable including defining generalism as “counting things in” (as opposed to failing or counting things out ) and  by “doing many things well” (again this positive spin is welcomed – and presumably could be tested.)

Thank you for this comment. We found a mix of CMO configurations that could apply to generalist choice and may apply to informing interventions.

To the Discussion we added about doing many things well:

“This could be strongly promoted as part of messaging within national campaigns.”

The authors are to be commended for keeping all the terms they used deliveratily neutral, and non-pejorative throughout. 

Thank you

The authors may find the recent popular book by David Epstein worthy reading:  Range, why generalists triumph in a specialised world.

Thank you for this reference, very thoughtful of you and this text is known to us. We added in other relevant literature to support the paper.
